# Association between Geriatric Nutritional Risk Index and Depression after Ischemic Stroke

**DOI:** 10.3390/nu14132698

**Published:** 2022-06-29

**Authors:** Jianian Hua, Jieyi Lu, Xiang Tang, Qi Fang

**Affiliations:** 1Department of Neurology, The First Affiliated Hospital of Soochow University, 899 Pinghai Road, Suzhou 215000, China; jnjnhua@foxmail.com (J.H.); goahead0523@163.com (J.L.); 2Medical College of Soochow University, 199 Renai Road, Suzhou 215123, China

**Keywords:** ischemic stroke, post-stroke depression, depression, geriatric nutritional risk index, malnutrition, China

## Abstract

Background: Malnutrition is associated with poor outcomes after stroke. However, the association between malnutrition and post-stroke depression (PSD) remains unelucidated. We aimed to explore the association between geriatric nutritional risk index (GNRI) and depression after ischemic stroke. Methods: In total, 344 patients with ischemic stroke were included in this analysis. The GNRI was calculated from serum albumin level, weight, and height at admission. Malnutrition was defined using the GNRI cutoff points. A lower GNRI score indicates an elevated nutritional risk. The outcome was depression, measured 14 days after ischemic stroke. Logistic regression models were used to estimate the association between the GNRI and risk of PSD. Results: A total of 22.9% developed PSD 14 days after stroke. The mean GNRI was 99.3 ± 6.0, and 53.8% of the patients had malnutrition. After adjusting for covariates, baseline malnutrition was not associated with risk of PSD (OR, 0.670; 95%CI, 0.370–1.213; *p* = 0.186). The restricted cubic splines revealed a U-shaped association between the GNRI and PSD. Compared to moderate GNRI, higher GNRI (OR, 2.368; 95%CI, 0.983–5.701; *p* = 0.085) or lower GNRI (OR, 2.226; 95%CI, 0.890–5.563; *p* = 0.087) did not significantly increase the risk of PSD. Conclusion: A low GNRI was not associated with an increased risk of depression after ischemic stroke.

## 1. Introduction

Stroke is associated with an increased incidence of morbidity and mortality worldwide. The number of new-onset strokes is 16.9 million annually and predicted to increase to 23 million by 2030 [1]. Depression is common after stroke. A meta-analysis published in 2005 reported a pooled prevalence of 33% and prevalence ranging from 14% to 41% [2,3,4]. Post-stroke depression (PSD) has a negative impact on cognitive recovery, functional outcomes, and quality of life. Patients with PSD sometimes find it harder to benefit from rehabilitation therapies. Hence, it is imperative to explore risk factors for PSD, in order to help identify patients at high risk and intervene early.

Malnutrition is associated with prognoses of cardiovascular and cerebrovascular disease [5,6,7]. Malnourished stroke survivors suffered from an increased risk of mortality and poorer functional outcomes, both in the short- and long-term, after stroke [8,9,10,11]. Patients at risk of malnutrition at admission tended to have a higher modified Rankin scale (mRS) score, poorer outcomes (mRS ≥ 3), and a higher mortality rate three months after stroke. Among patients with heart failure and acute coronary syndrome, malnutrition also correlates with increased all-cause mortality and adverse cardiovascular events [12,13].

Poor nutritional status was more frequently observed in participants with depression [14]. Conversely, malnutrition has been reported to be a risk factor for depression and increased depressive symptoms [6,15,16]. A study in Bangladesh reported that malnutrition was more frequent among depressed community-dwelling participants (56.0%) than non-depressed participants [17]. This study also suggested that malnourished participants had approximately three times higher risk of depression than well-nourished participants cross-sectionally. Although, studies have reported associations between malnutrition and depression among the general population. To the best of our knowledge, the relationship between malnutrition and depression among patients with ischemic stroke is yet to be reported.

Several screening tools have been used to assess malnutrition among patients with stroke, such as the malnutrition universal screening tool (MUST) and mini-nutritional assessment (MNA) [18]. Nevertheless, it might be difficult for patients with stroke to assess whether they have experienced recent weight loss or physical stress. The geriatric nutritional risk index (GNRI) is of growing interest for measuring malnutrition, and it is calculated using the following objective parameters: serum albumin level, height, and weight [19]. This index can be immediately obtained from medical records after hospitalization and is, therefore, more generalizable to general research. It has also been shown to correlate well with MNA in patients with stroke [4]. Recently, several studies have validated the predictive value of the GNRI for clinical outcomes in patients with malignant, heart, and renal diseases and stroke [8,12,20,21]. Lee et al. (2021) reported that lower GNRI scores were associated with a higher risk of post-stroke cognitive impairment [22] and lower cognitive scores in the Korean version of the mini-mental status examination and executive/activation domain. After stroke onset, cognitive impairment was highly correlated with depression at the same time point [23]. Hence, we hypothesized that lower GNRI scores might have a predictive role in PSD. Considering the lack of evidence on the association between malnutrition and PSD, exploring this association could contribute to the understanding of the pathogenesis, as well as potential treatments.

Therefore, the aim of our study was to investigate the relationship between GNRI and depression after ischemic stroke.

## 2. Materials and Methods

### 2.1. Study Participants

This was a retrospective study from a prospectively collected stroke neuropsychological database, carried out by the First Affiliated Hospital of Soochow University, and enrolled acute ischemic stroke patients who were consecutively admitted to the stroke unit. All protocols followed those outlined in the Declaration of Helsinki and were approved by the institutional review board of the participating hospital (IRB No. 2022-191). In reporting our study, we followed strengthening the reporting of observational studies in epidemiology (STROBE) guidelines [24]. Informed consent was obtained from all participants involved in the study.

Patients were included in the database if they were diagnosed with ischemic stroke by computed tomography (CT) or magnetic resonance imaging (MRI) and admitted within 3 days of symptom onset. The following patients were excluded: (a) a history of dementia, Parkinson’s disease, trauma, tumor, or other severe systemic diseases (*n* = 18); (b) a history of psychiatric or emotional disorders (*n* = 46); and (c) the inability to complete the assessment due to severe aphasia, apoplexy, or unconsciousness (*n* = 74).

From April 2021 to March 2022, 379 patients met the inclusion criteria for neuropsychiatric evaluation, while 24 patients were lost to follow-up or refused to offer information after discharge. In this study, we further excluded 11 participants without measurements of height, weight, or albumin levels, for a final sample of 344 patients with ischemic stroke (Figure 1).

### 2.2. Clinical Data

Fasting blood samples were routinely collected in the morning after hospital admission. Demographic characteristics and risk factors, including height and weight, were obtained within 6 h of admission.

The covariates included vascular risk factors (demographic characteristics and clinical history) and stroke features [25]. Heart disease consisted of atrial fibrillation and coronary disease. Based on a previous article, we chose to define the estimated glomerular filtration rate (eGFR) as a categorical variable with a cut-off point of 60 mL/min/1.73 m^2^ [26]. Stroke severity was assessed using the National Institutes of Health Stroke Scale (NIHSS) score at admission [27]. Stroke subtypes, according to the Trial of ORG 10,172 in Acute Stroke Treatment (TOAST) classification, were classified into four subtypes: large artery atherosclerosis (LAA), small vessel occlusion (SVO), cardioembolism (CE), and other types, including the other determined (OD) and undetermined (UD) [28]. Endovascular intervention was not considered, due to the lack of participants.

### 2.3. Assessment of Malnutrition and Post-Stroke Depression

Malnutrition was assessed using the GNRI score derived from the following formula: 1.489 × serum albumin (g/L) + 41.7 × actual weight (kg)/ideal weight. Ideal weight was calculated using the Lorenz formula: height (cm) − 100 − ([height (cm) − 150]/4) for men and height (cm) − 100 − ([height (cm) − 150]/2.5) for women. The actual weight/ideal weight ratio was regarded as 1, when the actual weight exceeded the ideal body weight. Patients were classified into two nutrition risk groups based on the GNRI: no malnutrition (GNRI ≥ 100) and malnutrition (GNRI < 100) [19].

The study outcome was depression evaluated using the 17-item Hamilton depression scale (HAMD-17) 14–21 days after stroke onset [3,29]. HAMD-17 has been translated into Chinese. Professional neurologists (J.Y. and X.T.) who assessed depression severity were unaware of the GNRI scores. PSD was diagnosed according to a HAMD-17 score ≥ 7 and the American Diagnostic and Statistical Manual of Mental Disorders Version 5 (DSM-5) [30,31].

### 2.4. Statistical Analysis

The baseline characteristics and GNRI scores were compared between the without-PSD and PSD groups and presented as mean ± standard deviation (SD), median (25th–75th percentile), or frequencies, as appropriate. For continuous variables, the Student’s *t*-test or Mann–Whitney U test was used to test the difference between groups, according to normality. The chi-square or Fisher’s exact tests were used to compare categorical variables (Table 1).

Logistic regression models were used to explore the association between GNRI and PSD. First, we compared the risk of PSD between the patients with and without malnutrition (Table 2). The odds ratios (ORs) and 95% confidence intervals (CIs) for the malnutrition group, compared with the group without malnutrition, were calculated. Second, we explored the linear associations between GNRI scores and PSD (Table 3) by calculating the ORs for the lower tertiles, compared to the highest tertile, and for each SD increment in GNRI scores to explore whether there was a linear association. Three logistic models were constructed: Model 1 was unadjusted; Model 2 was adjusted for age and sex; and Model 3 was adjusted for age, sex, education, clinical history (hypertension, diabetes mellitus, and heart disease), eGFR, admission NIHSS score, stroke subtype, and r-tPa treatment. To explore the pattern of the association between GNRI scores and PSD, restricted cubic splines with three knots (at the 5th, 50th, and 95th percentiles) adjusted for covariates included in model 3 were applied. Based on the pattern of the restricted cubic spline, differences in the risk of PSD between the groups with different degrees of malnutrition were calculated (Table 4).

All statistical analyses were performed using SAS version 9.4 (SAS Institute Inc., Cary, NC, USA) and 2-sided, with α = 0.05 being the threshold for statistical significance.

## 3. Results

The median age of the 344 enrolled participants was 65 (56–71) years. Most of the patients were male (67.5%). The mean GNRI score was 99.3 ± 6.0. According to the GNRI scores, 185 (53.8%) patients had some degree of malnutrition. Fourteen days after stroke, 79 (22.9%) patients developed depression.

Baseline characteristics, according to the development of depression, are presented in Table 1. Compared to individuals without PSD, those with PSD were more often female, had lower prevalence of diabetes mellitus, had a low eGFR score, had higher admission NIHSS scores, and had higher prevalence of OD + UD stroke subtype. There was no significant difference in baseline GNRI scores or the prevalence of malnutrition between the PSD and non-PSD groups.

The results of the associations between malnutrition (GNRI < 100) and PSD are shown in Table 2. Compared with patients without malnutrition, those with malnutrition showed a similar risk of PSD (OR: 0.696, 95%CI: 0.421–1.153, *p* = 0.160, for Model 1; OR: 0.720, 95%CI: 0.423–1.224, *p* = 0.225, for Model 2; OR: 0.670, 95%CI: 0.370–1.213, *p* = 0.186, for Model 3, respectively). As shown in Table 3, compared with those in the highest tertile of GNRI scores, those in the lower tertiles did not exhibit an increased risk of PSD. Furthermore, a 1-SD increment in the GNRI score was not associated with PSD.

The multiple-adjusted restricted spline regression models suggested that there might be a U-shaped association between GNRI scores and PSD (Figure 2). However, *p* for nonlinearity was not significant (*p* = 0.354). The GNRI scores could be categorized into four groups (without risk, ≥100; mild risk, 97.50–99.99; moderate risk, 83.50–97.49; and severe risk, <83.5). As shown in Figure 2, patients with a mild malnutrition risk seemed to have the lowest risk of PSD. Compared to patients with mild malnutrition risk, those without malnutrition risk (OR: 2.368, 95%CI: 0.983–5.701, *p* = 0.085, Model 3) or with moderate malnutrition risk (OR: 2.226, 95%CI: 0.890–5.563, *p* = 0.087, Model 3) did not have a significantly higher risk of PSD (Table 4).

In the sensitivity analysis, the aforementioned analyses were performed among patients with first-ever stroke and participants with mild stroke (NIHSS ≤ 4). Among patients with mild stroke, those without malnutrition risk had a higher PSD risk, compared to those with mild malnutrition risk in Model 3 (OR: 3.613, 95%CI: 1.050–12.434, *p* = 0.042) but not in Models 1 or 2 (Appendix A). Other sensitivity analyses yielded results similar to those presented in the main tables (Appendix A).

## 4. Discussion

In this study, we explored the relationship between the GNRI and depression in adults with ischemic stroke in China. Overall, we did not find a significant association between premorbid malnutrition, which was reflected by low GNRI scores, and the development of PSD 14 days after stroke.

### 4.1. GNRI as a Predictor for Stroke Outcome

A lower GNRI score, which reflects malnutrition, is a negative prognostic factor in various patient groups, including stroke patients. A Chinese study, including 1065 patients with first-ever stroke (mean age: 71 years), revealed that severe malnutrition (GNRI < 83.50) at admission increased the risk of mortality (HR = 3.641) during 8 years of follow-up after stroke, compared with patients with a GNRI score ≥ 100 [8]. A Japanese study of 540 stroke patients, with a mean age of 80 years, showed that a low GNRI (<92) at baseline and predicted poorer functional outcomes, as represented by the functional independence measure (FIM) gain [32]. Moreover, a recent Korean study, based on 344 patients with stroke (mean age: 62 years), indicated that malnutrition (GNRI < 98) at admission was associated with a higher risk of post-stroke cognitive impairment (OR = 2.04) and lower scores in the specific cognitive domains of executive/activation and language [22]. Compared to MNA and MUST, which require completing structured questionnaires, the GNRI is more convenient to achieve [18]. Most stroke units in China measure serum albumin levels upon admission. Researchers only need to measure height and weight. An electronic bed is required for patients who are unable to stand alone.

### 4.2. Malnutrition and Depression

There remains a lack of clinical evidence regarding whether baseline malnutrition can predict or aggravate depression after stroke. A study focusing on 307 ischemic stroke patients (mean age, 63 years) reported that low serum prealbumin levels at admission correlated with PSD one month after stroke onset [30]. Small-sample studies have observed that serum vitamin D and homocysteine levels could predict PSD [33,34]. Gu et al. studied 442 patients with stroke with a mean age of 62 years. Of the participants, 46% were vitamin D deficient or insufficient and had a higher risk of PSD one month after stroke (OR = 1.93) [34]. There were several randomized studies regarding the impact of nutritional support on outcomes after stroke. Nutritional supplementation included amino acids and vitamins. However, few published studies have reported the onset of depression as an outcome [35]. A randomized study in 2010, with 273 stroke patients (mean age: 63 years), suggested that treatment with B-group vitamin reduced the risk of depression (HR = 0.48) during a follow-up of 7.1 ± 2.1 years [36].

In contrast, depression is considered a risk factor for malnutrition among patients with stroke [35,37]. Stroke survivors with depression have less energy intake [38]. Furthermore, among the general population [6,7,15,39] and hospitalized patients [16,40], there was a correlation between poorer nutritional status and increased depressive symptoms. For example, a Greek survey of 2092 residents (74.97 ± 8.41 years) ascertained that those malnourished or at risk of malnutrition (based on MNA) had more depressive symptoms (measured by geriatric depression scale scores) [14]. Notably, most of these studies measured nutritional status and depression at the same time point, only achieving a cross-sectional association. In other words, the studies could only speculate that malnutrition was more frequently observed in participants with depression or malnourished participants had a higher prevalence of depression or more depressive symptoms. The cross-sectional design hindered causal inference. In this study, stroke participants were measured for GNRI at admission, at which time, the GNRI scores were close to the pre-stroke status. Meanwhile, participants with depression before stroke were excluded from the study. Therefore, our study setting may have reduced the disadvantages of the cross-sectional design. To achieve a relatively causal relationship between baseline GNRI and stroke outcome, future studies should explore longitudinal associations, which require repeated measurement data or follow-up data.

### 4.3. Advantage and Limitations

To our knowledge, this is the first report on the association between baseline nutritional status and the risk of PSD. This is also the first study to compare patients with and without PSD using the GNRI. Moreover, comprehensive information regarding potential covariates was collected at baseline. However, several disadvantages should be acknowledged when interpreting the results. First, patients with severe stroke have problems with consciousness or language function. The disability prevented us from performing neuropsychological tests. In the sensitivity analysis, we restricted our analyses to participants with only mild strokes and achieved similar results. Therefore, our conclusions can only be generalized to relatively healthy stroke patients. Second, the participants were all from China. Generalizability may be a concern. Third, we only included patients with ischemic stroke. The predictive effect of the GNRI in patients with intracerebral hemorrhage and subarachnoid hemorrhage is not clear. Fourth, only 4 of the 344 participants had a severe malnutrition risk. One article reported that only patients with stroke combined with severe malnutrition risk had a higher mortality risk, while mild and moderate malnutrition risk did not lead to adverse outcomes [8]. Future studies need to perform analyses among a much larger sample size, so that they can obtain a sufficient sample size in subgroups. Fifth, the exposure and outcome variables were recorded only once, limiting the exploration of the relationship between longitudinal changes in the GNRI and depression. The prevalence, severity, and risk factors of PSD may change during different periods of stroke, including the acute, subacute, and recovery periods (≥3 months) [41]. Future research could assess depression at three months, six months, and years after stroke. We suggest that there might be a bidirectional association between malnutrition and depression after stroke [42]. Sixth, the hospital-based design prevented us from evaluating the depressive scores before stroke onset. Confounding effects of pre-stroke depressive symptoms cannot be fully avoided. Future studies using cohorts with large sample sizes could learn the trends in depressive scores before and after stroke [43].

## 5. Conclusions

Our results do not support a significant relationship between baseline malnutrition and the risk of developing depression after stroke. Future research should explore the associations between longitudinal changes in malnutrition and depression after stroke using datasets with larger sample sizes and repeated measurements of nutritional status and neurophysiological tests.

## Figures and Tables

**Figure 1 nutrients-14-02698-f001:**
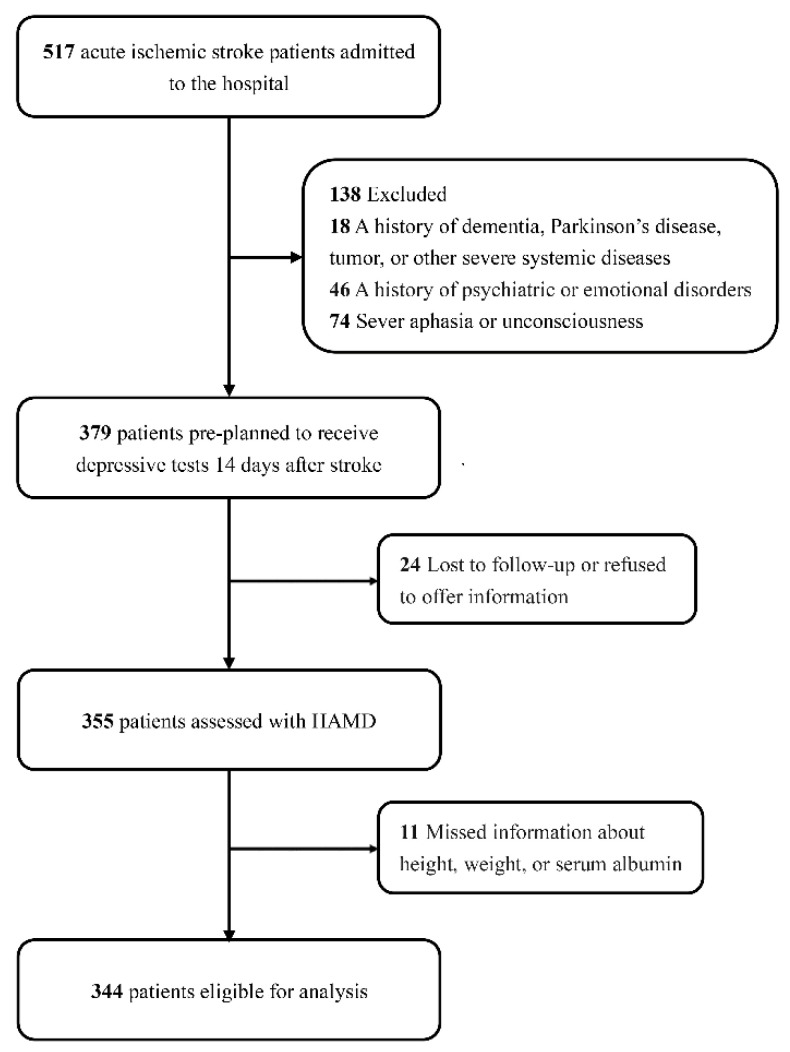
Study flow gram.

**Figure 2 nutrients-14-02698-f002:**
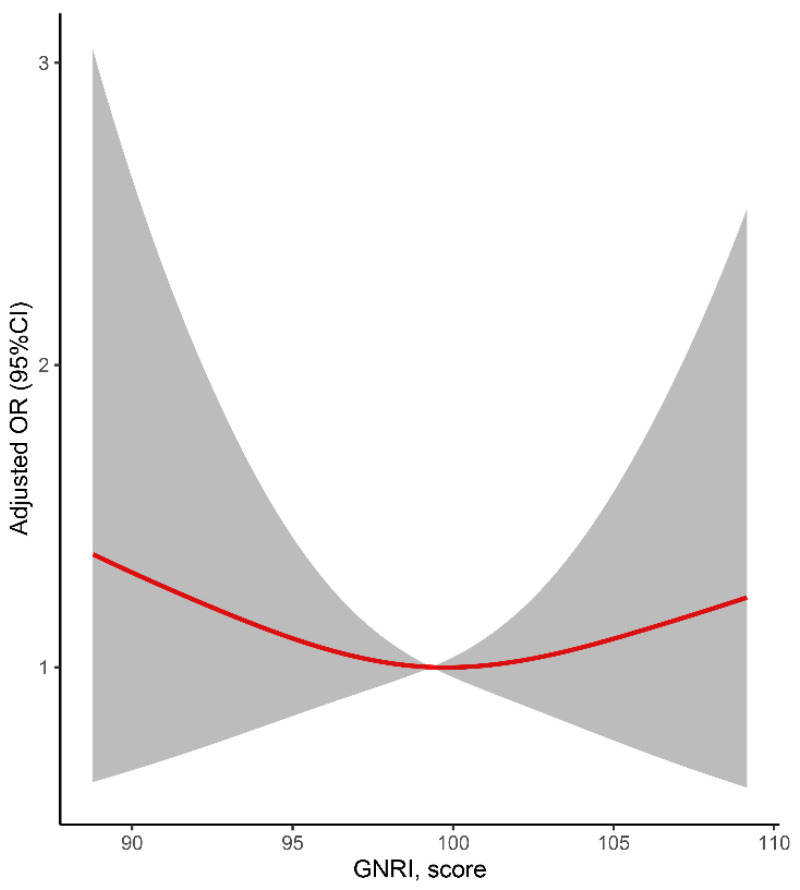
Relationship of GNRI and post-stroke depression in patients after ischemic stroke. Odds ratios and 95% confidence intervals derived from restricted cubic spline regression (*p* for nonlinearity = 0.354), with knots placed at the 5th, 50th, and 95th percentiles of the distribution of GNRI scores. The reference point was set at a GNRI value of 100. Odds ratios were adjusted for the covariates included in Model 3 in Table 2.

**Table 1 nutrients-14-02698-t001:** Baseline characteristics of participants, according to the presence of post-stroke depression.

	PSD (*n* = 79)	Without PSD (*n* = 265)	*p* Value
Malnutrition	37 (46.8)	148 (55.8)	0.198
GNRI	99.0 ± 6.4	99.4 ± 5.9	0.661
Age, years	64 (54–70)	65 (57–71)	0.782
Male	39 (49.4)	193 (72.8)	<0.001
Education			0.098
Illiteracy or primary school	33 (41.6)	105 (39.7)	
High school	31 (38.9)	139 (52.6)	
College or higher	15 (19.4)	21 (7.8)	
Hypertension	59 (74.7)	186 (70.2)	0.481
Diabetes mellitus	19 (24.1)	92 (34.7)	0.010
Heart disease	7 (8.9)	22 (8.3)	0.438
History of stroke	15 (18.9)	38 (14.3)	0.374
eGFR < 60 mL/min/1.73 m^2^	58 (73.4)	135 (50.9)	<0.001
Admission NIHSS score	3 (1–7)	2 (1–3)	0.003
Stroke subtype (TOAST)			
LAA	45 (56.7)	162 (61.1)	0.515
SVO	20 (25.3)	84 (31.7)	0.329
CE	6 (7.6)	9 (3.4)	0.121
OD + UD	8 (10.1)	10 (3.8)	0.040
r-tPa	11 (13.9)	52 (19.6)	0.320

Continuous variables were expressed as mean ± SD or median (interquartile range). Categorical variables are expressed as frequencies (percentages). Abbreviations: PSD, post-stroke depression; GNRI, geriatric nutritional risk index; eGFR, estimated glomerular filtration rate; NIHSS, National Institute of Health Stroke Scale; TOAST, Trial of ORG 10,172 in Acute Stroke Treatment; LAA, large artery atherosclerosis; SVO, small vessel occlusion; CE, cardioembolism; OD, other determined; UD, undetermined; r-tPA, recombinant tissue-type plasminogen activator.

**Table 2 nutrients-14-02698-t002:** Associations of malnutrition with risk of depression in patients after acute ischemic stroke.

	OR (95%CI)	*p* Value
Model 1	0.696 (0.421, 1.153)	0.160
Model 2	0.720 (0.423, 1.224)	0.225
Model 3	0.670 (0.370, 1.213)	0.186

Model 1: unadjusted model; Model 2: adjusted for age and sex; Model 3: adjusted for age, sex, education, hypertension, diabetes mellitus, heart disease, history of stroke, eGFR, admission NIHSS score, stroke subtype, and r-tPa treatment.

**Table 3 nutrients-14-02698-t003:** Associations of GNRI scores with risk of depression in patients after acute ischemic stroke.

	Tertile 1 (<100.7)	Tertile 2 (100.7–107.4)	Tertile 3 (≥107.4)	*P* Trend	Continuous (Per SD Increase)
Model 1	0.751 (0.257, 2.199)	0.778 (0.457, 1.324)	1.00	0.346	0.994 (0.733, 1.218)
Model 2	0.664 (0.204, 2.149)	0.820 (0.473, 1.971)	1.00	0.362	0.928 (0.701, 1.228)
Model 3	0.784 (0.484, 1.281)	0.631 (0.170, 2.334)	1.00	0.336	0.973 (0.709, 1.337)

Model 1: unadjusted model; Model 2: adjusted for age and sex; Model 3: adjusted for age, sex, education, hypertension, diabetes mellitus, heart disease, history of stroke, eGFR, admission NIHSS score, stroke subtype, and r-tPa treatment.

**Table 4 nutrients-14-02698-t004:** Associations of different levels of nutritional risk with risk of depression in patients after acute ischemic stroke.

Nutritional Risk	Model 1		Model 2		Model 3	
	OR (95%CI)	*p* Value	OR (95%CI)	*p* Value	OR (95%CI)	*p* Value
Without risk (*n* = 159)	1.974 (0.923, 4.224)	0.080	2.035 (0.933, 4.442)	0.074	2.368 (0.983, 5.701)	0.085
Mile risk (*n* = 65)	Reference		Reference		Reference	
Moderate risk (*n* = 116)	1.588 (0.712, 3.545)	0.259	1.893 (0.822, 4.359)	0.134	2.226 (0.890, 5.563)	0.087
Severe risk (*n* = 4)	Excluded		Excluded		Excluded	

Without risk: GNRI ≥ 100; mild risk: GNRI, 97.50–99.99; moderate risk: GNRI, 83.50–97.49; severe risk: GNRI < 83.50. Patients with severe risk were excluded, due to the small sample size. Model 1: unadjusted model; Model 2: adjusted for age and sex; Model 3: adjusted for age, sex, education, hypertension, diabetes mellitus, heart disease, history of stroke, admission NIHSS score, stroke subtype, eGFR, and r-tPa treatment.

## Data Availability

Data were entered into the stroke registration system of the First Affiliated Hospital of Soochow University (SR-FHSU). The data supporting the findings of this study are available from the corresponding author upon reasonable request.

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
