# Peer review of "Association between Geriatric Nutritional Risk Index and Depression after Ischemic Stroke"

_nutrients, 2022, doi:10.3390/nu14132698_

Round 1

Reviewer 1 Report

This manuscript has written well, however, this need to be revised just a few points. 

1) Introduction is not clear, why do you need to research this study? Could you describe more specificity?  

2) Authors showed that "This study was a retrospective study from a prospectively collected stroke~~~" in the present study.

However, selection of the patients are not clear, so could you show this using flow chart?

3) Are there any other study limitations in the present study?

Author Response

We thank the editors and reviewers for the criticisms, comments, and recommendations, all of which have helped us to significantly improve the quality of our manuscript.

This manuscript has written well, however, this need to be revised just a few points. 

  • Introduction is not clear, why do you need to research this study? Could you describe more specificity?  

Response: Thanks for the suggestion. We have made the introduction part more detailed. We added the following words:

Line 99 to Line 114 in clean version: “Malnutrition is associated with prognoses of cardiovascular and cerebrovascular disease [1-3]. Malnourished stroke survivors suffered from an increased risk of mortality and poorer functional outcomes both in the short and long term after stroke [4-7]. Patients at risk of malnutrition at admission tended to have a higher modified Rankin Scale (mRS) score, poorer outcomes (mRS 3), and a higher mortality rate three months after stroke. Among patients with heart failure and acute coronary syndrome, malnutrition also correlates with increased all-cause mortality and adverse cardiovascular events [8,9].

Poor nutritional status was more frequently observed in participants with depression [10]. Conversely, malnutrition has been reported to be a risk factor for depression and increased depressive symptoms [2,11,12]. A study in Bangladesh reported that malnutrition was more frequent among depressed community-dwelling participants (56.0%) than among non-depressed participants [13]. This study also suggested that malnourished participants had approximately three times higher risk of depression than well-nourished participants cross-sectionally. Although, studies have reported associations between malnutrition and depression among the general population. To the best of our knowledge, the relationship between malnutrition and depression among patients with ischemic stroke is yet to be reported.”

Line 124 to Line 127 in clean version:Lee et al. (2021) reported that lower GNRI scores were associated with a higher risk of post-stroke cognitive impairment [14] and lower cognitive scores in the Korean version of the Mini-Mental Status Examination and the Executive/Activation domain. After stroke onset, cognitive impairment was highly correlated with depression at the same time point [15].”

2) Authors showed that "This study was a retrospective study from a prospectively collected stroke~~~" in the present study.

However, selection of the patients are not clear, so could you show this using flow chart?

Response: Now we add a flow chart (Figure 1).

3) Are there any other study limitations in the present study?

Response: Thanks for your suggestion. We have added the limitations.
Line 286 to Line 300 in clean version: “Third, we only included patients with ischemic stroke. The predictive effect of the GNRI in patients with intracerebral hemorrhage and subarachnoid hemorrhage is not clear.” “Fifth, the exposure and outcome variables were recorded only once, limiting the exploration of the relationship between longitudinal changes in the GNRI and depression. The prevalence, severity, and risk factors of PSD may change during different periods of stroke, including acute, subacute, and recovery periods ( 3 months) [16]. Future research could assess depression at three months, six months, and years after stroke. We suggest that there might be a bidirectional association between malnutrition and depression after stroke [17]. Sixth, the hospital-based design prevented us from evaluating the depressive scores before stroke onset. Confounding effects of pre-stroke depressive symptoms cannot be fully avoided. Future studies using cohorts with large sample sizes could learn the trends in depressive scores before and after stroke[18].”

4) Are the methods adequately described? [can be improved]

Response: The method part is more detailed now.

5) Moderate English changes required 

Response: The English language had now been edited by a language company: Editage (www.editage.cn).

Reviewer 2 Report

In this study the authors explored the relationship between nutritional status evaluated by GNRI and post stroke depression. From both univariate and multivariate model the authors did not find significant relationship. Even though the study is interesting however I have some concerns:

a) Hamilton Depression Scale was evaluated after stroke and there are no data before the event

b) model 3 is adjusted for many variables. What were the criteria for choosing those variables? I would suggest to consider less variables.

c) other laboratory data would be of interest, for instance renal function, creatinine or GFR since albumin level may also be influenced by renal function. Also regarding comorbidities renal function it is not mentioned. 

Author Response

We thank the editors and reviewers for the criticisms, comments, and recommendations, all of which have helped us to significantly improve the quality of our manuscript.

In this study the authors explored the relationship between nutritional status evaluated by GNRI and post stroke depression. From both univariate and multivariate model the authors did not find significant relationship. Even though the study is interesting however I have some concerns:

  1. Hamilton Depression Scale was evaluated after stroke and there are no data before the event

Response: Thanks for your question. The hospital-based design prevented us from evaluating depressive scores before stroke onset. In our study and other similar studies, participants with a history of depression were excluded to avoid the confounding effect of pre-stroke depressive scores.

In “discussion-disadvantage” part, now we add “The hospital-based design prevented us from evaluating depressive scores before stroke onset. The confounding effects of pre-stroke depressive symptoms could not be fully avoided. Future studies using cohort with large sample size could learn the trends of depressive scores before and after stroke.”

  1. model 3 is adjusted for many variables. What were the criteria for choosing those variables? I would suggest to consider less variables.

Response: Thanks for your suggestion. 1) In the method part, now we added “Covariates included vascular risk factors (demographic characteristics and clinical history) and stroke features[19].” 2) Now we have reduced the confounders.

  1. c) other laboratory data would be of interest, for instance renal function, creatinine or GFR since albumin level may also be influenced by renal function. Also regarding comorbidities renal function it is not mentioned. 

Response: Thanks for your suggestion. We reduced the previous confounders and further controlled for eGFR. There was no difference in GFR between patients with and without depression while eGFR was calculated as a continuous variable. Based on a previous article, we chose to define eGFR as a categorical variable with a cut-off point of 60 mL/min/1.73 m2, achieving a P<0.001 for the difference between patients with and without depression[20]. After adjusting for the new covariates, the results remained similar.

Round 2

Reviewer 2 Report

The authors in overall addressed my questions / concerns.